# Decoding the Silent Majority: Inducing Belief Augmented Social Graph with Large Language Model for Response Forecasting

**Chenkai Sun, Jinning Li, Yi R. Fung, Hou Pong Chan,**
**Tarek Abdelzaher, ChengXiang Zhai, Heng Ji**
University of Illinois Urbana-Champaign
{chenkai5, czhai, hengji}@illinois.edu

## Abstract

Automatic response forecasting for news media plays a crucial role in enabling content producers to efficiently predict the impact of news releases and prevent unexpected negative outcomes such as social conflict and moral injury. To effectively forecast responses, it is essential to develop measures that leverage the social dynamics and contextual information surrounding individuals, especially in cases where explicit profiles or historical actions of the users are limited (referred to as lurkers). As shown in a previous study, 97% of all tweets are produced by only the most active 25% of users. However, existing approaches have limited exploration of how to best process and utilize these important features. To address this gap, we propose a novel framework, named SOCIALSENSE, that leverages a large language model to induce a belief-centered graph on top of an existent social network, along with graph-based propagation to capture social dynamics. We hypothesize that the induced graph that bridges the gap between distant users who share similar beliefs allows the model to effectively capture the response patterns. Our method surpasses existing state-of-the-art in experimental evaluations for both zero-shot and supervised settings, demonstrating its effectiveness in response forecasting. Moreover, the analysis reveals the framework's capability to effectively handle unseen user and lurker scenarios, further highlighting its robustness and practical applicability.

## 1 Introduction

*"Your beliefs become your thoughts. Your thoughts become your words. Your words become your actions."*
— Mahatma Gandhi

Automatic response forecasting (Figure 1) on receivers for news media is a burgeoning field of

The code is available at https://github.com/chenkaisun/SocialSense

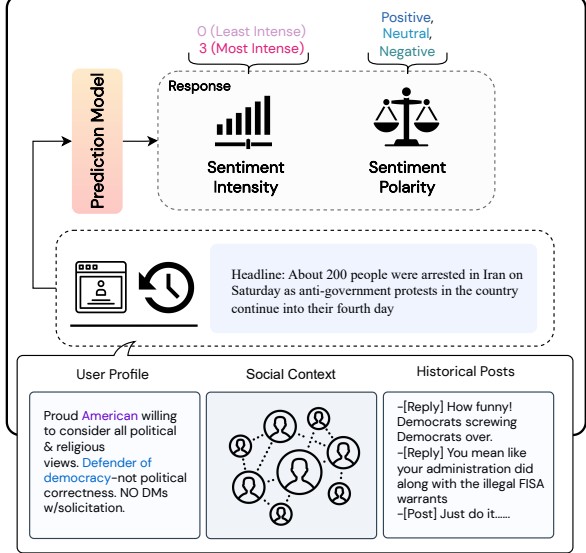

Figure 1: An example illustrating the task. The input consists of user attributes such as the profile and social context together with a news media message. The model is asked to predict response in multiple dimensions.

research that enables numerous influential applications, such as offering content producers a way to efficiently estimate the potential impact of their messages (aiding the prevention of unexpected negative outcomes) and supporting human writers in attaining their communication goals (Sun et al., 2023) for risk management. This direction is especially important nowadays as the proliferation of AI-generated misinformation, propaganda, and hate speech are becoming increasingly elusive to detection (Hsu and Thompson, 2023; Owen and Zahn, 2023). In this context, accurately forecasting the responses from different audiences or communities to news media messages becomes critical.

One of the primary challenges in personalized response forecasting lies in developing effective user representations. A crucial aspect to consider when representing a user is the integration of social dynamics (e.g., social interactions around a user) as well as their individual beliefs and interests. This becomes particularly relevant for users who lack

explicit profiles or historical activities (commonly referred to as lurkers). Previous efforts, however, have yet to explore the types of structural information that are helpful and how to best utilize such information (Lin and Chen, 2008; Giachanou et al., 2018; Yang et al., 2019; Wu et al., 2021).

During our preliminary analysis, we observed that users who share similar beliefs, specifically social values, are often situated in distant communities within the explicit social network. To provide further context, our findings reveal that a significant portion (over 44.6%) of users in the network data we collected for our experiment share beliefs with other users who are at least two hops away in the network. This emphasizes the importance of considering the connections between users with similar beliefs, even if they are not directly linked in the social network. Furthermore, previous research has indicated that user history plays a significant role in the model's performance. However, it is often directly utilized without processing in existing approaches, leading to the introduction of noise in the modeling process.

Motivated by these findings, we introduce **SO-CIALSENSE** (where Sense refers to the understanding and perception of social dynamics and behaviors within the online realm), a novel framework for modeling user beliefs and the social dynamics surrounding users in a social network. In this work, we conduct experiments using the **SOCIALSENSE** framework in the context of response forecasting. Our approach aims to capture the pattern of how "similar neighbors respond to similar news similarly". To harness the potential of network features, we curated a new user-user graph comprising 18k users from Twitter (the data will be anonymized when released), augmenting the original dataset (Sun et al., 2023). The **SOCIALSENSE** framework consists of three key stages: (1) inducing latent user personas using the Large Language Model (e.g., ChatGPT (Liu et al., 2023)), (2) building a belief-centered network on top of the existing social network, and (3) propagating information across multiple levels.

We demonstrate the effectiveness of our method through experiments on the dataset from Sun et al. (2023). Our results show that our framework outperforms existing baselines consistently across metrics in both zero-shot and fully-supervised settings. We further conduct a detailed analysis to address research questions concerning the model's general-

izability to unseen users and its predictive capabilities for lurkers. Our findings reveal two additional key insights: (1) the model performs exceptionally well in scenarios involving lurkers, outperforming the baseline by over 10% accuracy score in sentiment polarity forecasting, and, (2) compared to baseline approaches, the model exhibits consistently better generalization capabilities when applied to unseen users. Additionally, our analysis underscores the significance of various components within the belief-augmented social network, revealing that both the belief-centered graph and the user-news interaction network play vital roles in determining the network's overall performance.

## 2 Task Formulation

In the task of Response Forecasting on Personas for News Media, our objective is to predict how users will respond to news media messages. Specifically, we focus on analyzing the sentiment intensity and polarity of these responses. Formally, given a persona $\mathcal{P}$ (representing the user) and a news media message $\mathcal{M}$, our goal is to predict the persona's sentiment polarity $\phi_p$ (categorized as either *Positive*, *Negative*, or *Neutral*) and intensity $\phi_{int}$ (measured on a scale of 0 to 3) of the persona's response. We frame this task as a multi-class prediction problem.

## 3 SOCIALSENSE

To accurately predict individuals' responses, it is crucial to develop an effective user representation that captures their personas. While previous studies have utilized user profiles and historical data to model individuals' interests with reasonable accuracy, there is a significant oversight regarding the behavior of a large number of internet users who are passive participants, commonly referred to as lurkers. This phenomenon is exemplified by statistics showing that only 25% of highly active users generate 97% of the content on Twitter (McClain et al., 2021). Consequently, the sparse historical data available for lurkers makes it challenging to infer their responses reliably. To address this issue, a social network-based approach can be employed to leverage users' social connections, gathering information from their neighbors. However, it is important to question whether relying solely on social networks is sufficient.

In this work, we introduce a novel perspective by borrowing the concept of belief and defining it in terms of social values. By considering so-

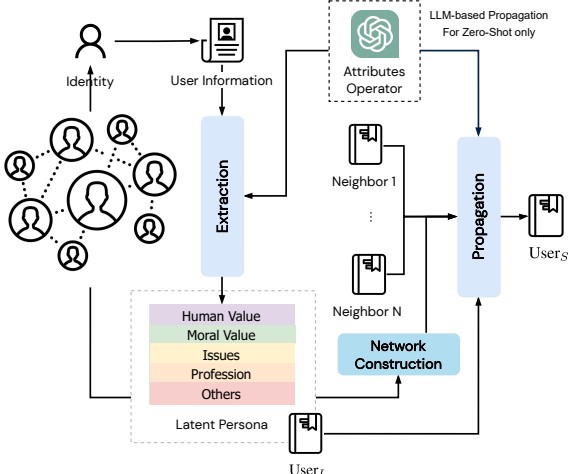

Figure 2: The figure illustrates our framework. In the first stage, we use an LLM to extract latent persona from the user's profile and historical posts. These moral and human value attributes from the latent personas, combined with the social network and news media messages, collectively shape the belief-augmented social network. Graph-based propagation is then used to update user representation. In the zero-shot setting, the LLM itself also assumes the role of an information propagator that combines information from neighbors (more details in Section 3.4).

cial values, which encompass human values and moral values, we capture individuals' deeply held convictions, principles, and ethical standards that significantly shape their perspectives, behaviors, and responses within a social context. Our preliminary analysis reveals that individuals who share beliefs are often distantly connected, beyond residing in the same community. Specifically, we found that over 44.6% of users in our collected network data share beliefs with others who are at least two hops away in the network. This finding highlights the potential value of bridging these distant users and incorporating their beliefs as valuable features in response forecasting.

In this study, we present **SOCIALSENSE** (Figure 2), an innovative framework for modeling user beliefs and the social dynamics within a social network by automatically curating a belief-centered social network using a Large Language Model (e.g., ChatGPT). Our approach consists of three stages: (1) extracting latent personas using a Large Language Model, (2) constructing a belief-centered network on top of the existing social network, and (3) information propagation. In addition to the supervised method, we further explore how to achieve zero-shot prediction with social networks by simulating graph propagation with SOCIAL PROMPT.

## 3.1 Unmasking Latent Persona with Large Language Model

Although the user's past posts can provide insights into their interests, they often contain noise that makes them challenging for models to consume. For instance, they may describe life events without providing context, such as "*@user Waited all day next to phone. Just got a msg...*". Furthermore, relying solely on raw historical data discourages explainability in response forecasting since past utterances are influenced by a person's internal beliefs rather than being the sole determinant of their future response.

In recent months, the Large Language Models (LLMs), particularly ChatGPT, have been shown to surpass human annotators in various tasks given their effective training techniques and access to vast amounts of pretraining data (Gilardi et al., 2023). This breakthrough presents unprecedented opportunities in analyzing users comprehensively without being scoped by previously established research. For the first time, we leverage a large language model (specifically, ChatGPT in our experiment) to extract users' internal beliefs and construct beliefs suitable for downstream consumption.

In this initial stage of our framework, we design a prompt $P_l$ that enables us to extract latent information not available anywhere online. This includes dimensions such as human values, moral values, views on entities and issues, professions, and more. The prompt we have developed is shown in the Appendix. We refer to the latent persona extracted from the LLM for a user as $User_L$. In other words,

$$User_L = \mathbf{LLM}(\text{profile}, \text{history}, P_l) \qquad (1)$$

## 3.2 Belief-Augmented Social Network

To capture social interactions and bridge distant communities, our approach incorporates both existing and induced social information to construct a network that focuses on modeling users' beliefs.

Our graph can be formally defined as follows: it comprises three sets of nodes, namely $\mathcal{V}^M$ representing the news media messages, $\mathcal{V}^U$ representing the users, and $\mathcal{V}^B$ representing a fixed set of belief nodes. The graph consists of three types of edges: $\mathcal{E}^I$, $\mathcal{E}^F$, and $\mathcal{E}^B$. For each edge $(u, m) \in \mathcal{E}^I$, where $u \in \mathcal{V}^U$ and $m \in \mathcal{V}^M$, it indicates that user $u$ has interacted with the news media message $m$. For

each edge $(u_1, u_2) \in \mathcal{E}^F$, where $u_1, u_2 \in \mathcal{V}^U$, it signifies that user $u_1$ follows user $u_2$. Lastly, for each edge $(u, b) \in \mathcal{E}^B$, where $u \in \mathcal{V}^U$ and $b \in \mathcal{V}^B$, it denotes that user $u$ believes in the value represented by node $b$. An illustrative example sub-graph of the network is shown in Figure 3.

**Social Relation Network**   The first layer of our network consists of the user-user social network, where edges from User $a$ to $b$ indicate that User $a$ follows User $b$. This network captures the interests of users and the relationships between users.

**User-Media Interactions**   The second component of our network comprises news nodes and response edges indicating the users in the network have responded to these news nodes in the dataset. This feature offers two advantages. Firstly, it serves as a representation of users' interests. Secondly, it facilitates the connection of users who are geographically distant in the network but might share interests in news topics, thus enabling the expansion of the set of potentially reliable neighbors for any user we would like to predict.

**Belief-Centered Graph**   Lastly, we introduce belief nodes, composed of moral and human values (principles that guide behaviors) from the Latent Personas.

MORAL VALUES: Moral values are derived from a set of principles that guide individuals or societies in determining what is right or wrong, good or bad, and desirable or undesirable. We define the set of Moral Values based on the Moral Foundations Theory (Graham et al., 2018), which includes Care/Harm, Fairness/Cheating, Loyalty/Betrayal, Authority/Subversion, and Purity/Degradation.

HUMAN VALUES: Human values are defined based on the Schwartz Theory of Basic Values (Schwartz, 1992), encompassing Conformity, Tradition, Security, Power, Achievement, Hedonism, Stimulation, Self-Direction, Universalism, and Benevolence. These values represent desirable goals in human life that guide the selection or evaluation of actions and policies.

Building upon the network from the previous stage, we establish connections between users and their associated values in an undirected manner. This connection type offers two key benefits. Firstly, it introduces shortcuts between users who share similar beliefs or mindsets, facilitating the propagation of information across distant nodes. Secondly, it allows the prediction results of user responses to potentially be attributed to the belief

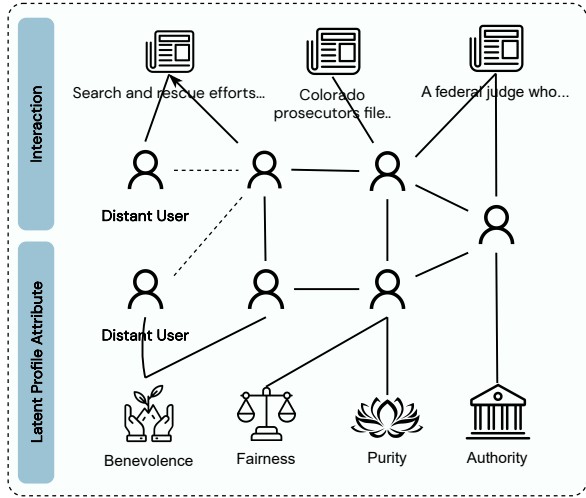

Figure 3: An example illustrating a snapshot of the belief-centered social network. The latent persona attributes serve as a bridge between (potentially distant) users who share values. The arrow on the top left refers to the response we aim to forecast.

nodes (instead of past utterances), thereby enhancing the explainability of the process.

### 3.3   Information Propagation

Given the constructed belief graph, we utilize a Graph Neural Network (GNN) (Zhou et al., 2020) to propagate information and learn an updated user representation, enabling us to infer user responses.

**Node Initialization**   To train the GNN, we first need to initialize the node representations. For user nodes $\mathcal{V}^U$, we leverage a Pretrained Language Model (PLM) such as DeBERTa (He et al., 2020) to encode the user's profile and history, yielding a $d$-dimensional dense vector $\mathbf{u}$. Similarly, we initialize media nodes $\mathcal{V}^M$ by encoding the news headline message by the PLM, obtaining vector $\mathbf{m}$. The embeddings for the fixed set of belief nodes $\mathcal{V}^B$, $\mathbf{b}$, are initialized by random vectors.

**Graph Propagation**   We consider response forecasting as a reasoning process over the connections among news media, user, and belief nodes in the social graph. Leveraging the social homophily phenomenon, we posit that the constructed social ties lead to the formation of communities reflecting similarities and differences in beliefs, both within and across communities. To capture the interactions across different types of graph components, we employ a Heterogeneous Graph Transformer (HGT) (Hu et al., 2020), which was inspired by the architecture of the classic Transformer (Vaswani et al., 2017). Unlike homogeneous GNNs, HGT

effectively handles different edge and node types as separate meta paths, facilitating the learning of user representations from various types of contextual nodes.

Upon obtaining the updated user representations from HGT, we concatenate them with the news embeddings. The resulting vector is passed through an MLP layer followed by a softmax activation function for classification. The model is trained using cross-entropy loss, where the labels are sentiment intensity/polarity.

### 3.4 Zero-Shot Prediction by Simulating Propagation with Social Prompts

To forecast responses in a zero-shot fashion, one approach involves directly feeding user profiles, historical data, and news headlines into large language models like ChatGPT. However, this approach lacks the inclusion of the user's social network and encounters challenges when dealing with lurkers who have limited background information. As demonstrated in the experiment section, including social context provides a clear advantage in response forecasting. In this section, we introduce the concept of SOCIAL PROMPT to simulate information propagation in the supervised setting.

**Neighborhood Filtering** To aggregate information, one needs to select information from neighbors. Since language models have a limited context window and a user typically has hundreds of followers/followings, we filter the set of neighbors by ranking the neighbors based on their influence on the user's opinion. In our design, we utilize the concept of authority from the persuasion techniques (Braca and Dondio, 2023), using the number of followers a neighbor has to determine their level of influence. We select the top-$K$ neighbors $\mathcal{N}^K$ as the filtered set to represent the social context of the central user.

**Aggregation and Prediction** Given the latent user personas attributes, $\text{User}_L^n$ extracted for each neighbor $n \in \mathcal{N}^K$ of central node $c$, extracted from Section 3.1 for each neighbor, and the filtered neighborhood from the previous step, we construct a prompt $P_s$ (shown in the Appendix) that allows the LLM to produce a socially aware persona $\text{User}_S$. Finally, we design a prediction prompt $P_p$, which utilizes both $\text{User}_L$ and $\text{User}_S$ of the central node to make predictions. Formally,

$$\mathcal{R} = \text{LLM}(P_p, U_L^c, \text{LLM}(P_s, \{U_L^n\}^{n \in \mathcal{N}^K}))$$

(2)

where U abbreviates User, $U^c$ indicates the current central user, and $\mathcal{R}$ indicates the prediction results.

## 4 Experiment

### 4.1 Data Construction

We use the dataset from (Sun et al., 2023) (denoted as RFPN) as the base for evaluation. The dataset consists of 13.3k responses from 8.4k users to 3.8k news headlines collected from Twitter. More details are shown in the Appendix.

**Network Data** To test SOCIALSENSE, we curate a social network using the official Twitter API[1]. We initialize the network with the users in RFPN $X_s$. We collect all the users that each user $u \in X_s$ follows and denote them as $X_t$. We then select the top 10000 followed accounts from $X_t \cup X_s$ as the most influential nodes, and denote them $X_f$. Lastly, we merge the top influencers with the original user set $X_s$ into the final set $\mathcal{V}^U = X_f \cup X_s$. Our final graph consists of $18,634$ users and $1,744,664$ edges.

### 4.2 Experimental Setup

**Evaluation Metrics** We evaluate the prediction of sentiment intensity using the Spearman and Pearson correlation, which are denoted as $r_s$ and $r$, respectively. For the classification of sentiment polarity, we evaluate with the Micro-F1 score (or equivalently accuracy in the multi-class case) and Macro-F1 score, denoted as MiF1 and MaF1.

**Baselines** We conduct a comparative analysis of SOCIALSENSE with several baseline models, including *DeBERTa* (He et al., 2020) (upon which our node initialization is based) and *RoBERTa* (Liu et al., 2019b), which are state-of-the-art pretrained language models known for their performance across various downstream tasks like sentiment analysis and information extraction. Additionally, we compare our approach with the *InfoVGAE* model (Li et al., 2022), a state-of-the-art graph representation learning model specifically designed for social polarity detection. InfoVGAE constructs a graph that captures the edges between users and news articles to learn informative node embeddings. We extend this model by incorporating user-user edges and also an additional two-layer MLP classifier head to adapt it for our supervised tasks. Furthermore, we include two naive baselines, namely *Random* and *Majority*. The *Ran-*

---

[1] https://developer.twitter.com/en/docs/twitter-api

*dom* baseline makes predictions randomly, while the *Majority* baseline follows the majority label. These baselines serve as simple reference points for comparison. Lastly, we compare our response forecasting results with *ChatGPT*, a state-of-the-art zero-shot instruction-following large language model (LLM) (Yang et al., 2023). To predict the sentiment intensity and polarity using ChatGPT, we use the prompt $P_p$ from Section 3.4 that incorporates the user profile, user history, and the news media message as the input. We leverage the official OpenAPI with the `gpt-3.5-turbo` model[2] for sentiment prediction.

To illustrate the effectiveness of SOCIAL PROMPTS (Section 3.4), we compare three models: baseline ChatGPT, ChatGPT$_L$, and SocialSense$_{Zero}$. In ChatGPT$_L$, we incorporate the latent persona User$_L$ from Section 3.1, while in SocialSense$_{Zero}$, we leverage the aggregated social context User$_S$ generated by SOCIAL PROMPT in addition to User$_L$ (Section 3.4). We use $K = 25$ for SOCIAL PROMPT. Similarly, we utilize the prompt $P_p$ for response prediction. The detailed prompts can be found in the Appendix.

**Implementation and Environments**  Our neural models are implemented using Pytorch (Paszke et al., 2019) and Huggingface Transformers (Wolf et al., 2020). The intensity label in the dataset follows the definition in the SemEval-2018 Task 1[3] (Mohammad et al., 2018), where the sign is also considered during evaluation. More implementation details and discussions of reproducibility and hyperparameters can be found in the Appendix.

### 4.3 Results Discussion

We conduct an evaluation of the proposed SO-CIALSENSE model and the baseline models introduced in Section 4.2 for the supervised response forecasting task. The evaluation results are presented in Table 1. While the state-of-the-art models demonstrate competitive performance, SO-CIALSENSE *outperforms* all other models across all evaluation metrics consistently. Although Chat-GPT is designed and proven effective for zero-shot instruction-following text generation, we observe that its performance in sentiment forecasting of responses is comparatively limited, yielding lower scores compared to the other supervised models.

[2]https://platform.openai.com/docs/api-reference/models
[3]https://competitions.codalab.org/competitions/17751

| Method | $\phi_{int}$ (%) | | $\phi_p$ (%) | |
| --- | --- | --- | --- | --- |
| | $r_s$ | $r$ | MiF1 | MaF1 |
| Majority | - | - | 43.41 | 20.18 |
| Random | 0.62 | 0.41 | 35.51 | 30.55 |
| ChatGPT | 43.80 | 44.15 | 58.61 | 48.67 |
| DeBERTa | 50.81 | 50.58 | 64.77 | 59.30 |
| RoBERTa | 52.09 | 53.00 | 65.26 | 59.02 |
| InfoVGAE | 58.61 | 58.37 | 67.46 | 60.05 |
| SocialSense | **61.82** | **61.98** | **70.45** | **65.71** |
| w/o belief | 59.92 | 60.06 | 66.80 | 59.70 |
| w/o user-news | 55.43 | 55.35 | 66.51 | 61.96 |
| w/o profile | 59.94 | 60.01 | 64.49 | 59.04 |
| w/o history | 57.60 | 57.29 | 67.95 | 62.89 |
| w/ random init | 58.25 | 58.40 | 61.79 | 56.44 |

Table 1: Response forecasting results. We report the Spearman and Pearson correlations for the forecasting of sentiment intensity, as well as Micro F1 and Macro F1 scores for the sentiment polarity prediction. The best overall performance is in bold. Our framework outperforms the baselines consistently.

This highlights that *the task can not be fully addressed by a zero-shot model alone.*

On the other hand, the RoBERTa and DeBERTa models, despite being smaller pre-trained models, exhibit relatively better correlation and F1 scores after fine-tuning for our response prediction task on news articles. However, these models only utilize textual information from news articles and user profiles, disregarding potential interaction patterns and shared beliefs among users. This explains why their correlations and F1 scores are, on average, $10.28\%$ and $5.99\%$ lower than those achieved by the proposed SOCIALSENSE framework. Additionally, the graph-based InfoVGAE model achieves higher scores compared to the text-based DeBERTa and RoBERTa baselines, highlighting the significance of graph-structured data in enhancing response forecasting performance. However, the evaluation metrics of the InfoVGAE model remain lower than those of SOCIALSENSE. While the InfoV-GAE model constructs a graph primarily based on user-user and user-news interaction edges, SO-CIALSENSE goes a step further by inducing and integrating additional belief nodes and edges. This novel approach results in a heterogeneous graph that forges connections among users who share similar perspectives and ideologies, thereby facilitating the learning of intricate social dynamics and bolstering the model's predictive capabilities.

### 4.4 Ablation Study

We conduct an ablation study on different components of SOCIALSENSE to evaluate their impact on

performance. The results are presented in Table 1.

**Belief-Centered Graph** To assess the effectiveness of the Belief-Centered Graph in Section 3.2, we conduct an experiment where we removed the belief nodes from the graph, including the nodes representing moral values and human values. This leads to a decrease of $1.91\%$ in correlations and $4.83\%$ in F1 scores. These findings support our hypothesis that incorporating belief nodes is effective in modeling the shared beliefs and values among users. By including belief nodes, we enable the graph learning framework to capture the association between the underlying principles and moral frameworks that guide users' behaviors and response patterns.

**User-News Edges** In this experiment, we exclude the user-news edges while constructing the belief-augmented heterogeneous graph. The results show that modeling the user-news interaction as edges results in an improvement of up to $6.63\%$ in correlation metrics for sentiment intensity prediction. This indicates that modeling users' interests and historical interactions with media is crucial for accurately predicting sentiment intensity.

**User Profile and Historical Posts** The ablation study reveals the important roles of user profile data and historical post data in response forecasting. Excluding user profile data leads to a drop of $1.93\%$ and $6.32\%$ on average in the respective tasks, emphasizing its significance in predicting sentiment polarity. Removing historical post data results in a decrease of approximately $4.45\%$ in correlations and $2.66\%$ in F1 scores for sentiment polarity prediction. These findings highlight the importance of both data types, with profile data influencing intensity prediction more and historical data affecting polarity prediction more.

**Node Initialization** Instead of using the text representations of users' profiles and historical posts, we randomly initialize the node features. This results in a decrease of $3.57\%$ in correlations and a significant decrease of $8.97\%$ in F1 scores for polarity classification, emphasizing the significance of text features in predicting sentiment polarity.

### 4.5 Zero-Shot Evaluation

In addition to supervised response forecasting, we also evaluate our framework under the zero-shot setting (Section 3.4). The results are presented in Table 2. Based on the higher scores attained by ChatGPT$_L$, it is evident that the in-

| Method | $\phi_{int}$ (%) | | $\phi_p$ (%) | |
|---|---|---|---|---|
| | $r_s$ | $r$ | MiF1 | MaF1 |
| ChatGPT | 43.8 | 44.15 | 58.61 | 48.67 |
| ChatGPT$_L$ | 44.43 | 44.76 | 59.77 | 48.69 |
| SocialSense$_{Zero}$ | **46.64** | **47.22** | **60.54** | **51.30** |

Table 2: The above Zero-Shot Response forecasting results highlight that the SOCIAL PROMPT from Section 3.4 consistently offers an advantage.

| Method | $\phi_{int}$ (%) | | $\phi_p$ (%) | |
|---|---|---|---|---|
| | $r_s$ | $r$ | MiF1 | MaF1 |
| Case Study: Lurker Users | | | | |
| DeBERTa | 39.58 | 36.72 | 59.20 | 51.98 |
| RoBERTa | 43.21 | 41.67 | 60.81 | 52.74 |
| InfoVGAE | 37.37 | 36.60 | 61.34 | 47.61 |
| SocialSense | **50.30** | **53.57** | **71.01** | **63.88** |
| Case Study: Unseen Users | | | | |
| DeBERTa | 41.72 | 39.32 | 55.56 | 48.80 |
| RoBERTa | 38.06 | 35.71 | 55.20 | 47.99 |
| InfoVGAE | 36.08 | 35.06 | 56.27 | 47.86 |
| SocialSense | **44.40** | **44.27** | **62.55** | **55.37** |

Table 3: The case studies for Lurker and Unseen User Scenarios demonstrate that our framework exhibits significantly improved generalization capabilities when the user is unseen or has limited background context.

clusion of latent structured persona information indeed aids the model in comprehending the user more effectively. Furthermore, our model, SO-CIALSENSE$_{Zero}$, achieves the highest scores consistently across all metrics. This demonstrates the efficacy of our method for zero-shot social context learning and provides compelling evidence that even in the zero-shot setting, social context plays a crucial role in response forecasting.

### 4.6 Evaluation on Lurker and Unseen User Scenarios

We evaluate the performance of proposed models and baselines on the task of response forecasting for lurker users, who are characterized as users with only a small amount of historical posts. In the experiment, we define the lurkers as the users with less than 50 historical responses (less than 85% of the users in the dataset), and the scenario consequently contains 745 test samples. The scores are shown in Table 3. Compared to the previous evaluation results in Table 1, we observe that the overall evaluation scores for all the models are significantly lower. This can be attributed to the fact that lurkers have a much smaller background context, making response prediction more challeng-

ing. The lurker case is especially difficult for those baselines relying heavily on historical responses. In this challenging scenario, **SOCIALSENSE** not only achieves significantly higher scores than others in all of the metrics but also maintains its performance on the polarity measures. Specifically, the advantage of our proposed model over DeBERTa and RoBERTa expands from 5.99% to 11.26% in terms of F1 scores for sentiment polarity prediction. These results demonstrate that even in cases where user textual information is extremely limited, our framework can still accurately infer responses, showcasing the robustness of our method. Furthermore, it is worth noting that the intensity score was noticeably lower compared to the regular setting, indicating that predicting the intensity of responses becomes more challenging when historical information is limited. We conduct further evaluation of the proposed model and baselines on unseen users, which refers to the responders who only appear in the evaluation dataset. This case study on unseen users provides insights into the generalization of the models. The evaluation results are presented in Table 3. The results indicate that the unseen user scenario presents a more challenging task compared to previous settings. Moreover, **SOCIALSENSE** demonstrates significantly higher performance across all metrics compared to other baselines. This outcome underscores the framework's ability to effectively generalize to unseen users, likely attributed to its robust modeling of the social network and encoding of relationships between users.

## 5 Related Work

Existing research has focused on predicting the individual-level response using additional textual features as well as deep neural networks (DNN) (Lin and Chen, 2008; Artzi et al., 2012; Li et al., 2019; Wang et al., 2020). However, these existing methods neglected the important information about users' personas as well as the modeling of graph-structured interactions among users with the social items. Another line of related works formulates the response forecasting as text-level generation task (Yang et al., 2019; Wu et al., 2021; Lu et al., 2022; Wang et al., 2021). However, these lack a quantitative measure for analyzing the response (such as in the sentiment dimensions), limiting their applicability in downstream tasks like sentiment prediction on impact evaluation of

news (Sun et al., 2023). In contrast, we propose a novel framework that leverages large language models to induce the graph structure and integrates disentangled social values to forecast responses, whether in a supervised or zero-shot manner. Our work demonstrates that effectively modeling the social context and beliefs of users provides a clear advantage in the social media response forecast task. This can ultimately benefit various downstream applications such as assisting fine-grained claim frame extraction (Gangi Reddy et al., 2022) and situation understanding (Reddy et al., 2023).

In the field of Social-NLP, related research has focused on applying NLP techniques, large language models (LLM), and prompting strategies to model, analyze, and understand text data generated in social contexts. For instance, progress has been made in misinformation detection (Fung et al., 2021; Wu et al., 2022; Huang et al., 2023b) and correction (Huang et al., 2023a), propaganda identification (Martino et al., 2020; Oliinyk et al., 2020; Yoosuf and Yang, 2019), stance detection (Zhang et al., 2023), ideology classification (Kulkarni et al., 2018; Kannangara, 2018), LM detoxification (Han et al., 2023), norms grounding (Fung et al., 2023), popularity tracking (He et al., 2016; Chan and King, 2018), and sentiment analysis (Araci, 2019; Liu et al., 2012; Azzouza et al., 2020). The emergence of advanced decoder language models like ChatGPT has led to extensive research on prompting techniques and their application across various NLP tasks (Zhou et al., 2022; Kojima et al., 2022; Zhao et al., 2021; Diao et al., 2023; Sun et al., 2022). Indeed, experiments have shown that ChatGPT even outperforms crowd workers in certain annotation tasks (Gilardi et al., 2023). However, when it comes to social tasks like response forecasting, relying solely on large-scale models without taking into account the social context and users' personas may not yield optimal performance (Li et al., 2023). Our experiments demonstrate that incorporating social context in the prompt consistently enhances the LLM's performance, as showcased in our simulation of information propagation using large language models.

## 6 Conclusions and Future Work

In conclusion, we present **SOCIALSENSE**, a framework that utilizes a belief-centered graph, induced by a large language model, to enable automatic response forecasting for news media. Our framework

operates on the premise that connecting distant users in social networks facilitates the modeling of implicit communities based on shared beliefs. Through comprehensive evaluations, we demonstrate the superior performance of our framework compared to existing methods, particularly in handling lurker and unseen user scenarios. We also highlight the importance of the different components within the framework. In future research, it would be valuable to explore the application of belief-augmented social networks in other domains and to develop an effective social prompting strategy for general-purpose applications. Furthermore, it is worth investigating how response forecasting models can adapt efficiently to dynamically evolving data, especially given the swift changes observed in real-world social media platforms (de Barros et al., 2023; Cheang et al., 2023).

## Limitations

While the proposed **SOCIALSENSE** framework demonstrates promising results in response forecasting, there are limitations to consider. Firstly, the performance of the model heavily relies on the quality and availability of social network data. In scenarios where these sources are extremely limited or noisy, the model's predictive capabilities may be compromised. Additionally, the generalizability of the framework to different domains and cultural contexts needs to be further explored and evaluated.

## Ethics Statements

The primary objective of this study is to enable content producers to predict the impact of news releases, thereby mitigating the risk of unforeseen negative consequences such as social conflict and moral injury. By providing a stronger and more robust framework for forecasting responses, we aim to contribute to the creation of a safer online environment. In our process of collecting the network data using Twitter API, we strictly adhere to the Twitter API's Terms of Use[4]. As part of our commitment to responsible data handling, we will release only an anonymized version of the network data when making the code repository publicly available.

---

[4] https://developer.twitter.com/en/developer-terms/agreement-and-policy

## Acknowledgement

This research is based upon work supported in part by U.S. DARPA INCAS Program No. HR001121C0165. The views and conclusions contained herein are those of the authors and should not be interpreted as necessarily representing the official policies, either expressed or implied, of DARPA, or the U.S. Government. The U.S. Government is authorized to reproduce and distribute reprints for governmental purposes notwithstanding any copyright annotation therein.

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

# A Appendix

## A.1 Implementation Details

We implement the training framework using the 4.8.2 version of Huggingface Transformer library[5](Wolf et al., 2020). For the graph model implementation in Section 3.3, we use the 2.0.3 version of PyG[6]. The hyperparameters for the experiment are shown in Table 4 and the ones not listed in the table are set to be default values from the transformer library. We use RAdam (Liu et al., 2019a) as the optimizer. We perform greedy hyperparameter search on the gnn_layer from {1,2,3}, learning rate from {5e-5, 1e-4, 5e-4, 1e-3}, # attention heads from {2, 4, 6, 8}, activation from {tanh, relu}, # epochs from {350, 1000}, and node

---

[5]https://github.com/huggingface/transformers
[6]https://pytorch-geometric.readthedocs.io/en/latest/index.html

dimensions from {128, 256}. We perform our experiments on a single NVIDIA RTX A6000 48 GB. Our model consists of 10, 484, 424 tuning parameters and it takes less than 30 minutes to fine-tune.

## A.2 Analysis of Belief Data

We perform additional analysis on the belief data. Specifically, we show the distribution of the belief data (Figure 4), for which the moral value of care is dominant among the users. We have also segregated the model's performance in sentiment prediction based on the users' belief values and show it in Table 6. Empirical results indicate that the model is more accurate when predicting sentiments for users characterized by universalism and degradation. Conversely, the model finds it challenging to predict sentiments for users in the categories of security and stimulation. We further sampled 50 ChatGPT extraction results from user histories and distributed them among three human raters to assess the accuracy of the extracted profiles. These raters are graduate students who qualified through an initial quiz comprising eight samples. On evaluation, the raters assigned an average score of 3.9 out of 5 for accuracy. While not flawless, these extracted beliefs play a significant role in boosting the model's performance. Such finding indicates that refining the ChatGPT extraction process could potentially lead to enhanced performance outcomes.

## A.3 Prompts Templates

We show all prompts used in the work in Figure 5, Figure 6, Figure 7, Figure 8, and Figure 9. They represent $P_l$, $P_s$, $P_p$ for the baseline ChatGPT, $P_p$ for ChatGPT$_L$, and $P_p$ for SocialSense$_{Zero}$ respectively.

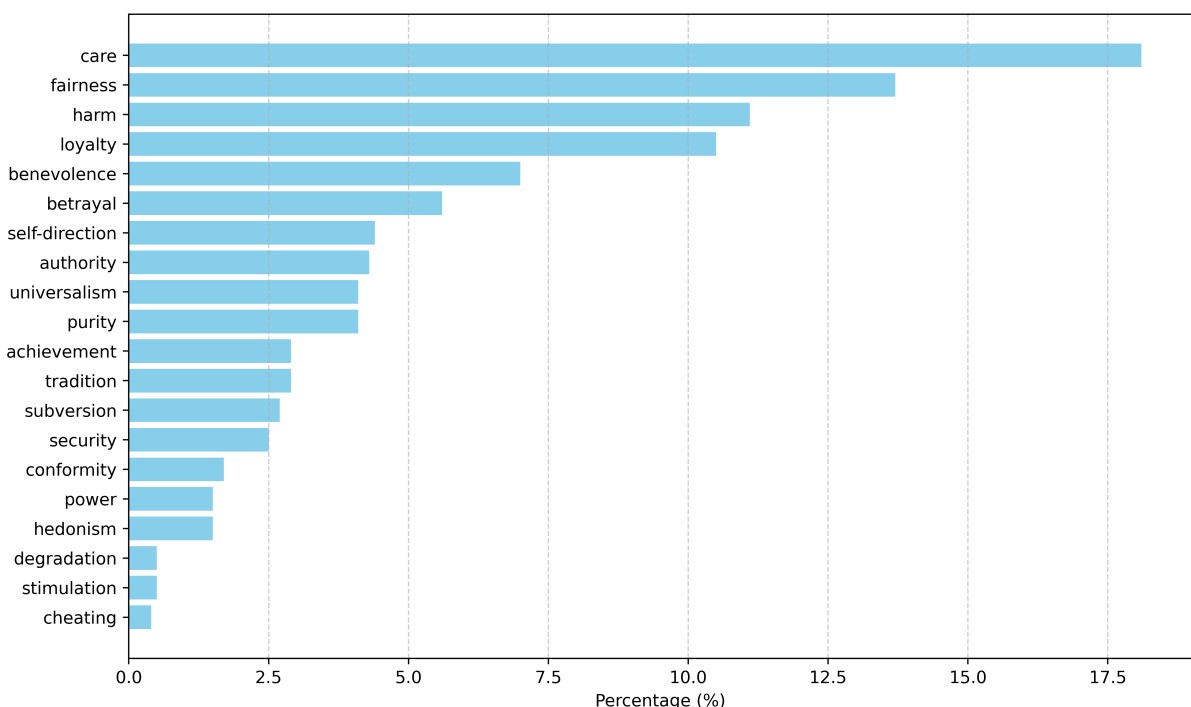

Figure 4: Distribution of belief values

```
Describe the Twitter user (based on its profile and history) by filling in the categories below COMPLETELY and
COMPREHENSIVELY, without losing information. Do not rename the categories. Separate by newline:
Human Values (i.e., what human values does the user likely have. Choose from "Conformity", "Tradition", "Security",
"Power", "Achievement", "Hedonism", "Stimulation", "Self-Direction", "Universalism", "Benevolence". These are based
on the basic theory of human values. Separate by commas),
Moral Values (i.e., what moral values does the user likely have. choose from "authority", "betrayal", "care",
"cheating", "degradation", "fairness", "harm", "loyalty", "purity", "subversion". Separate by commas),
Ideologies (i.e., choose from political ideologies. Separate by commas),
Possessions (i.e., what the user owns),
Interested topics (i.e., what topics is the user interested in. This can be inferred from the profile and history.
Separate by commas),
Interested issues (or events) and the user's stance toward each issue (i.e., fill the template exactly "Support:
<issues> ; Neutral: <issues> ; Against: <issues>". Fill <issues> with "None" if you are not sure for that label),
Interested entities and the user's stance toward each entity (i.e., fill the template exactly "Support: <entities> ;
Neutral: <entities> ; Against: <entities>". Fill <entities> with "None" if you are not sure for that label),
Profession (e.g., jobs, specialty. Separate by commas),
Social roles (e.g., mother, father. Separate by commas),
Other Notes.

Answer the categories concisely and comprehensively. If a category cannot be answered, fill in the exact word "None"
for the corresponding category.

I will iteratively provide the user profile and user historical posts. Here is the information,

user twitter profile: '{profile}',

user historical tweets (separated by semi-colons): '{history}'
```

Figure 5: Prompt template $P_l$ used for extracting user latent profile User$_L$ in Section 3.1. The input consists of user profile text and concatenated user historical posts. The output contains categories filled answers.

```
Assume there is a user called User_0, and there are many accounts (who are also users) User_0 follows, and these
accounts form the community around User_0. I will provide a descriptions for each of these accounts. Do the
following question: Summarize the neighborhood context (in terms of dominant information) around User_0. That is,
describe the neighborhood community around this User_0 (This is used to represent the User_0's belief and social
context). Also guess and describe the User_0 itself using the information (of its neighborhood) provided.

In other words, filling in the categories below (using template like <category>:<answer>) (using as few words as
possible) but completely and comprehensively, without losing information. Separate by newline:
Dominant Human Values (i.e., choosing from "Conformity", "Tradition", "Security", "Power", "Achievement",
"Hedonism", "Stimulation", "Self-Direction", "Universalism", "Benevolence". These are based on the basic theory of
human values. Otherwise simply choose "None" if not sure. Separate by commas),
Dominant Moral Values (i.e., choosing from "authority", "betrayal", "care", "cheating", "degradation", "fairness",
"harm", "loyalty", "purity", "subversion". Otherwise simply choose "None" if not sure. Separate by commas),
Dominant Ideologies (i.e., choosing from political ideologies, or None, separated by commas),
Dominant Interested topics (i.e., Choose None if you cannot answer. Separate by commas),
Dominant issues (or events) and the user's stance toward each issue (i.e., fill the template exactly "Support:
<issues> ; Neutral: <issues> ; Against: <issues>". Fill <issues> with "None" if you are not sure for that label),
Dominant Interested entities and the user's stance toward each entity (i.e., fill the template exactly "Support:
<entities> ; Neutral: <entities> ; Against: <entities>". Fill <entities> with "None" if you are not sure for that
label),
Dominant Professions (e.g., jobs, specialty. Separate by commas),
Description of the User_0 (i.e., using the information of its neighborhood provided),
Other Notes.

Answer the categories concisely and comprehensively. If there is no clear dominant trend in a category, fill in the
exact word "None" for the corresponding category.

==========

Here are the neighbors' information:

{list of latent profiles from each neighbor}
```

Figure 6: Prompt template $P_s$ used for aggregating neighbor information in Section 3.4. It takes a list of latent profiles from neighbors, where the latent profiles are output from $P_l$.

```
Predict response from the user to the news headline in terms of exact comment words (i.e., what would user reply in
comment), sentiment polarity (i.e., Positive, Neutral, Negative), and sentiment intensity (integer scaled between 0-3
inclusively where 0 means no intensity and 3 means the most intense). Note: when sentiment polarity is neutral, the
sentiment intensity should be 0.

In other words, filling the <answer> in the categories below. Separate by newline:
Comment: <answer>
Sentiment Polarity: <answer>
Sentiment Intensity: <answer>

I will iteratively provide the news headline, user profile, and user historical posts. Here is the information,

[news headline]: '{post}',

[user profile]: '{profile}',

[user historical posts]: '{history}'.
```

Figure 7: Prompt template $P_p$ used for predicting responses given only news message, user profile text, and concatenated user historical posts as input. It is used for evaluating the baseline ChatGPT.

```
Predict response from the user to the news headline in terms of exact comment words (i.e., what would user reply in
comment), sentiment polarity (i.e., Positive, Neutral, Negative), and sentiment intensity (integer scaled between 0-3
inclusively where 0 means no intensity and 3 means the most intense). Note: when sentiment polarity is neutral, the
sentiment intensity should be 0.

In other words, filling the <answer> in the categories below. Separate by newline:
Comment: <answer>
Sentiment Polarity: <answer>
Sentiment Intensity: <answer>

I will iteratively provide the news headline, user profile, user historical posts, user structurized profile, and
user social context. Here is the information,

[news headline]: '{post}',

[user profile]: '{profile}',

[user historical posts]: '{history}'.

[user structurized profile]: '{latent profile of the user from P_l}'
```

Figure 8: Prompt template $P_p$ using user latent profile in addition to input Figure 7. It is used for evaluating ChatGPT$_L$.

```
Predict response from the user to the news headline in terms of exact comment words (i.e., what would user reply in
comment), sentiment polarity (i.e., Positive, Neutral, Negative), and sentiment intensity (integer scaled between 0-3
inclusively where 0 means no intensity and 3 means the most intense). Note: when sentiment polarity is neutral, the
sentiment intensity should be 0.

In other words, filling the <answer> in the categories below. Separate by newline:
Comment: <answer>
Sentiment Polarity: <answer>
Sentiment Intensity: <answer>

I will iteratively provide the news headline, user profile, user historical posts, user structurized profile
(inferred by AI), and user social context (inferred by AI) which describes the community around the user (User_0
indicates the user itself). Here is the information,

[news headline]: '{post}',

[user profile]: '{profile}',

[user historical posts]: '{history}'.

[user structurized profile]: '{latent profile of the user from P_l}'

[user social context (community)]: '{aggregated social context from P_s}'
```

Figure 9: Prompt template $P_p$ using aggregated social context User$_S$ from Section 3.4 in addition to input Figure 8. It is used for evaluating SocialSense$_{\text{Zero}}$ in the experiment.

| Name | Value |
|---|---|
| seed | 42 |
| learning rate | 5e-4 |
| batch size | 1 |
| weight decay | 5e-4 |
| RAdam epsilon | 1e-8 |
| RAdam betas | (0.9, 0.999) |
| scheduler | linear |
| warmup ratio (for scheduler) | 0.06 |
| number of epochs | 1000 |
| patience (for early stop) | 300 |
| # gnn layers | 3 |
| # attention head | 4 |
| activation | ReLU |
| dropout | 0.2 |
| node dimensions | 128 |

Table 4: Hyperparameters

| Belief Value | MiF1 | MaF1 |
|---|---|---|
| conformity | 62.96 | 59.35 |
| tradition | 57.14 | 48.29 |
| security | 50.00 | 43.42 |
| power | 66.67 | 60.84 |
| achievement | 69.23 | 57.91 |
| hedonism | 56.25 | 46.03 |
| stimulation | 33.33 | 16.67 |
| self-direction | 59.68 | 48.61 |
| universalism | 73.02 | 64.04 |
| benevolence | 62.04 | 50.80 |
| authority | 62.50 | 56.40 |
| betrayal | 60.61 | 50.09 |
| care | 62.81 | 52.19 |
| cheating | 75.00 | 42.86 |
| degradation | 87.50 | 46.67 |
| fairness | 64.04 | 56.22 |
| harm | 66.29 | 54.46 |
| loyalty | 68.28 | 60.17 |
| purity | 70.59 | 60.56 |

Table 6: Performance segmented by users' belief values.

| Split | Train | Dev. | Test |
|---|---|---|---|
| # Samples | 10,977 | 1,341 | 1,039 |
| # Headlines | 3,561 | 1,065 | 843 |
| # Users | 7,243 | 1,206 | 961 |
| Avg # Profile Tokens | 10.75 | 11.02 | 10.50 |
| Avg # Response Tokens | 12.33 | 12.2 | 11.87 |
| Avg # Headline Tokens | 19.79 | 19.82 | 19.72 |

Table 5: Summary statistics for the original dataset.