# OpenReview forum: "Decoding the Silent Majority: Inducing Belief Augmented Social Graph with Large Language Model for Response Forecasting"
_EMNLP/2023/Conference — EMNLP 2023 Main_

### Official Review · Reviewer_9BLt · 2023-08-02

**Typos Grammar Style And Presentation Improvements:** 1. N_{L} should be N_{K} in Line 360?
**Soundness:** 3

**Excitement:**

3: Ambivalent: It has merits (e.g., it reports state-of-the-art results, the idea is nice), but there are key weaknesses (e.g., it describes incremental work), and it can significantly benefit from another round of revision. However, I won't object to accepting it if my co-reviewers champion it.

**Paper Topic And Main Contributions:**

This paper aims to predict the personalized user response to a specific news item, where a comprehensive user representation plays a vital role. Though existing works have employed user profiles, history posts, and social networks to enhance the social contexts of users, the authors claim that they are ineffective in dealing with lurkers. To this end, the authors propose to incorporate user beliefs and offer the SOCIALSENSE framework. The framework utilizes LLMs to infer the social values of user beliefs and augment the social network with a belief-centered graph. A heterogeneous graph transformer is then adopted to learn user and news representations from the augmented graph and infer the personalized user response. Experimental results demonstrate the effectiveness of the proposed framework, especially for lurkers and unseen users.

**Questions For The Authors:**

Q1: Now that LLMs cannot make accurate predictions of user responses, how can they make accurate predictions of user beliefs for lurkers? How you analyzed the prediction quality of user beliefs?

Q2: Compared to the baselines, what's the number of parameters of your proposed framework?  How much time does it cost to fine-tune your model?


**Reasons To Accept:**

A1. The idea of incorporating user beliefs for response forecasting is reasonable and can enhance the explainability of the predictions.

A2. The performance gain is significant in both zero-shot and supervised settings. The analysis experiment supports the claim that involving user beliefs facilitates the inference of lurkers.

A3. Utilizing LLMs to infer human values sounds interesting and worth further exploration. The proposed social prompt is also noteworthy, which mimics the feature aggregation in GNNs in a prompting way.

**Reasons To Reject:**

R1. Though LLMs show some merits in serving as data annotation tools, the outcomes need further verification, especially in such an unsupervised way. Analysis of the annotation results should be included.

R2. It is unclear why LLMs can provide convincing predictions of user beliefs for lurkers. If a big error happens at this stage, it will propagate to the following processing stages. This is also why R1 is important.

R3: The introduced value nodes might bring noise and increase the complexity. As I understand, almost all the users are 2-hop away via the "user-value-user" path. By adopting a 3-layer GNN, the users will gather information from a large number of 2-hop neighbors, which might bring noise. And the graph is much denser, resulting in increased complexity.

R4: The model details are missing, e.g., the choice of meta-paths when applying HGT.

**Reproducibility:**

3: Could reproduce the results with some difficulty. The settings of parameters are underspecified or subjectively determined; the training/evaluation data are not widely available.

**Reviewer Confidence:**

4: Quite sure. I tried to check the important points carefully. It's unlikely, though conceivable, that I missed something that should affect my ratings.

---

> ### Author Rebuttal · Authors · 2023-08-29
>
> We thank Reviewer 9BLt for the insightful comments. We also appreciate the reviewer’s positive feedback on the significance of the performance gain and the novelty of using LLM for unmasking latent personas. We address the comments as follows.
>
> **[== analysis of LLM annotation and error propagation of noisy input ==]**
>
> We understand the reviewer’s concern regarding the quality of LLM annotation. We performed a validation study on the accuracy of 50 ChatGPT extraction results (distributed to three human raters) and we found that on average the raters gave a score of 3.9/5 in terms of accuracy. We have also made an analysis of belief data. Specifically, we show the distribution of the belief data in which the moral value care is dominant among the users:
>
> | Belief Value      | % |
> | --------------- | ------ |
> |care  | 18.1%         |
> |fairness |  13.7%     |
> |harm  | 11.1%         |
> |loyalty  | 10.5%      |
> |benevolence |  7.0%   |
> |betrayal  | 5.6%      |
> |self-direction  | 4.4%|
> |authority  | 4.3%     |
> |universalism |  4.1%  |
> |purity  | 4.1%        |
> |achievement |  2.9%   |
> |tradition  | 2.9%     |
> |subversion  | 2.7%    |
> |security  | 2.5%      |
> |conformity  | 1.7%    |
> |power  | 1.5%         |
> |hedonism |  1.5%      |
> |degradation  | 0.5%   |
> |stimulation  | 0.5%   |
> |cheating  | 0.4%      |
>
> We will include these results in the final version.
>
> In our work, we employed the latent personas from LLM as extra nodes to improve the model. Our empirical results show that the introduced nodes are indeed effective, implying that the quality of the current annotation does contribute to the performance positively. For lurker cases where history is limited, we hypothesize that user-user might be a balancing indicator of user social context. We agree that it is important to improve the LLM annotation since the method relies on its quality. We will discuss how to improve LLM annotation in the revision.
>
> We agree with the reviewer that some of the results from LLM might be subjected to errors since it’s not a perfect classifier. It is therefore important to analyze error propagation to understand the effect of noise better. In the final revision, we will incorporate an error analysis that quantifies how often negative performance is attributed to errors in the LLM annotation.
>
> Regarding Question 1, we also would like to note that the classification of user beliefs and forecasting of user response might have some differences. In the former case, a model is asked to perform classification on user history (e.g., a user who produces posts about praying for people who suffer from disaster has a moral value of care), while the latter is relatively harder as it is about forecasting reactions given often insufficient context (e.g., the user has scarce data or never discussed topics related to the message), which we find is sometimes even challenging for humans.
>
>
>
> **[== noise of introduced value nodes and model detail ==]**
>
> We understand the reviewer’s concern that the value nodes might introduce noise. For the user-value-user path, two users are both connected by a value node if they have common values (out of 20 defined ones), forming implicit communities. In our parameter searching, we find the 3-layer to be relatively more effective (but the number of attention heads is also important) and we will include the analysis of the effect of the number of gnn layers in the final version. We also agree that solving the graph complexity issue certainly would shed light on improving the performance. One potential solution is to introduce weight on different meta paths. For instance, if we hypothesize one of the paths might introduce noise, we can investigate whether setting a lower weight can help performance.
>
> For meta-paths, we have user *is associated with* value, user *follows* user, and user *responds to* news in our graph construction. Our model consists of 10,484,424 tuning parameters and it takes less than 30 minutes to fine-tune. We will include the details in our revision.
>
>
>
> **[== writing and presentation ==]**
>
> We thank the reviewer for pointing out the symbol error and we will correct it in our final version.

---

### Official Review · Reviewer_EsDk · 2023-08-02

**Soundness:** 4

**Excitement:**

4: Strong: This paper deepens the understanding of some phenomenon or lowers the barriers to an existing research direction.

**Paper Topic And Main Contributions:**

This paper proposes a new framework for predicting the sentiment intensity and polarity of twitter users. It creates a graph of users, posts, and beliefs, and includes user-user edges, user-post edges, and user-belief edges. The belief vertices are composed of moral and human values derived from latent personas of users. The latent personas are extracted using ChatGPT. Information is propagated through the constructed graph using a Heterogeneous Graph Transformer. The results are evaluated using correlation coefficients and F1 scores.

This paper proposes a framework using a large language model to generate a belief-centered graph to augment the existing social network. However, insights about the belief networks and clusters of users associated with certain beliefs are not discussed.

**Questions For The Authors:**

1. Are certain beliefs harder to detect than others? Is there any performance gap associated with groups of different beliefs?
2. How is the ground truth sentiment intensity and polarity calculated based on the user and message?

**Reasons To Accept:**

1. The problem statement is well-defined and the method is clearly described.
2. The results for the Lurker and Unseen User scenarios are quite strong, which big improvements shown over the baselines.
3. The empirical evaluation includes several baselines and ablation studies, which helps to indicate how well the model performs.

**Reasons To Reject:**

1. More insights about the interaction between users and beliefs or the distribution of identified beliefs are not included. As this is one of the main contributions of this paper, it would be beneficial to show some analysis of what the beliefs data looks like in this network.
2. The results of the proposed model are incrementally better than the baselines for response forecasting.
3. The framework is only evaluated on the task of sentiment intensity and polarity prediction. It would be helpful to include other evaluation tasks as well.

**Reproducibility:**

4: Could mostly reproduce the results, but there may be some variation because of sample variance or minor variations in their interpretation of the protocol or method.

**Reviewer Confidence:**

3: Pretty sure, but there's a chance I missed something. Although I have a good feel for this area in general, I did not carefully check the paper's details, e.g., the math, experimental design, or novelty.

---

> ### Author Rebuttal · Authors · 2023-08-29
>
> We thank Reviewer EsDk for the helpful and detailed comments. If given more space, we will be able to address the comments and improve our current draft.
>
> **[==  the level of performance improvement  ==]**
>
>
> We understand the reviewer's concern regarding the magnitude of the performance improvement as shown in some parts of the tables. It's worth noting, however, that our proposed method consistently outperforms all baselines across a variety of experimental scenarios.
>
>
> In this work, we hypothesize that including network structure and belief nodes can address the issue of forecasting the response of lurkers. Through evaluation, we find that not only the issue is largely alleviated (with up to 11.9% increase in scores shown in Table 4), but the method also works well in unseen user cases from Table 4 (with up to 6.57% increase) and full data case (with 5.66% increase) from Table 2, all with consistent improvement over the best baseline. To further substantiate these results, we will add a significance test in our final version.
>
>
> We would like to also note that the perception of performance improvement level might also depend on the difficulty of the task and domain. In our data analysis stage, both the social scientists consulted for this work and the authors concur that the task is inherently challenging; unlike many text-level classification tasks, our model often has to operate with limited user data, making predictions difficult even for human experts. To build upon this, future research will explore augmenting user context with multimodal information, such as images and videos, and will investigate the potential benefits of adding more anchor nodes.
>
>
> **[== introducing more evaluation tasks ==]**
>
> We appreciate the reviewer’s suggestion on further tasks the method can be evaluated upon. We believe one of such tasks is measuring whether the user might be offended, which allows the method to be useful for more application scenarios. We will include the evaluation in our next revision.
>
>
> **[== analysis of belief data ==]**
>
> We have made further analysis of belief data. Specifically, we show the distribution of the belief data, for which the moral value care is dominant among the users:
>
> | Belief Value      | % |
> | --------------- | ------ |
> |care  | 18.1%         |
> |fairness |  13.7%     |
> |harm  | 11.1%         |
> |loyalty  | 10.5%      |
> |benevolence |  7.0%   |
> |betrayal  | 5.6%      |
> |self-direction  | 4.4%|
> |authority  | 4.3%     |
> |universalism |  4.1%  |
> |purity  | 4.1%        |
> |achievement |  2.9%   |
> |tradition  | 2.9%     |
> |subversion  | 2.7%    |
> |security  | 2.5%      |
> |conformity  | 1.7%    |
> |power  | 1.5%         |
> |hedonism |  1.5%      |
> |degradation  | 0.5%   |
> |stimulation  | 0.5%   |
> |cheating  | 0.4%      |
>
>
> We have also segregated the model's performance in sentiment prediction based on the users' belief values. Empirical results indicate that the model is more accurate when predicting sentiments for users characterized by universalism and degradation. Conversely, the model finds it challenging to predict sentiments for users in the categories of security and stimulation. We believe one natural direction for model improvement is to enhance persona extraction quality by, for example, taking ensemble predictions from multiple models. On the other hand, since belief values do not always hold equal significance in predicting responses to specific types of news messages, we could introduce a weighted attention mechanism to better prioritize certain belief values during the graph propagation process.
>
> | Belief Value      | MiF1 | MaF1 |
> | --------------- | ------ | ------- |
> | conformity  | 62.96  | 59.35     |
> | tradition  | 57.14  | 48.29      |
> | security  | 50.0  | 43.42        |
> | power  | 66.67  | 60.84          |
> | achievement | 69.23  | 57.91     |
> | hedonism  | 56.25  | 46.03       |
> | stimulation  | 33.33 |  16.67    |
> | self-direction  | 59.68  | 48.61 |
> | universalism  | 73.02  | 64.04   |
> | benevolence  | 62.04  | 50.8     |
> | authority  | 62.5 |  56.4        |
> | betrayal  | 60.61 |  50.09       |
> | care |  62.81  | 52.19           |
> | cheating  | 75.0  | 42.86        |
> | degradation  | 87.5  | 46.67     |
> | fairness |  64.04 |  56.22       |
> | harm  | 66.29  | 54.46           |
> | loyalty |  68.28  | 60.17        |
> | purity  | 70.59 |  60.56         |
>
> We will include the results in the final version.
>
>
> **[== ground truth labels ==]**
>
> We would like to note that the ground truth sentiment intensity and polarity labels have been annotated in the previous dataset (Sun et al., 2023). In the dataset, the author annotated labels on each response from a user to a message. In our modeling, we train the model to perform multi-class prediction.

---

### Official Review · Reviewer_HaTs · 2023-08-11

**Soundness:** 4

**Excitement:**

4: Strong: This paper deepens the understanding of some phenomenon or lowers the barriers to an existing research direction.

**Missing References:**

None.

**Paper Topic And Main Contributions:**

The paper introduces a novel approach to predict user responses to news media messages by leveraging ChatGPT to extract user beliefs. The primary contribution lies in the construction of a belief-centered social network through a GNN.
This network captures the patterns of how similar neighbors respond to similar news.
And the experiements validate its improved generalization, accuracy, and robustness.

**Questions For The Authors:**

Were there any challenges in dealing with label inconsistencies in ChatGPT? For example, generating novel labels beyond the predefined set. If so, how were these challenges addressed?

Is there a possibility of label leakage or user overlap between the training data (collected by you) and the testing data (Sun et al., 2023)? If such overlap exists, how might it impact the evaluation results and the generalizability of the proposed approach?


Could you please provide insight into the rationale behind your choice of belief features and latent personas derived from MORAL VALUES and HUMAN VALUES? Additionally, do you have plans or ideas for extending this list of belief sources to further enhance the effectiveness of your approach?

**Reasons To Accept:**

The paper's novelty is commendable, particularly in its utilization of LLMs for practical applications. The combination of a supervised GNN and LLMs demonstrates an effective solution for complex tasks.
The ablation study strongly supports the argument that supervised learning along with LLM knowledge is more effective than pure supervised learning  or zero-shot approaches.

**Reasons To Reject:**

The writing style could benefit from being more concise and focused.

**Reproducibility:**

4: Could mostly reproduce the results, but there may be some variation because of sample variance or minor variations in their interpretation of the protocol or method.

**Reviewer Confidence:**

3: Pretty sure, but there's a chance I missed something. Although I have a good feel for this area in general, I did not carefully check the paper's details, e.g., the math, experimental design, or novelty.

**Typos Grammar Style And Presentation Improvements:**

The structure of the paper needs improvement. The abstract, introduction, and methods sections seem to overlap in content. Section 2 is too brief, while Section 3 takes a while to introduce the exact proposed method. Balancing the sections and their content could enhance the overall flow of the paper.

Consider including a figure to illustrate the zero-shot and supervised settings. Figure 2 could be enhanced for clarity by simplifying its structure. Reducing the number of arrow directions and arranging them in a more logical sequence would aid comprehension.

---

> ### Author Rebuttal · Authors · 2023-08-29
>
> We thank Reviewer HaTs for the constructive and detailed comments. We appreciate the reviewer’s positive feedback on the novelty and effectiveness of the approaches. If given more space, we will be able to address all comments.
>
>
> **[== writing style and presentation ==]**
>
> First, we thank the reviewer for the thoughtful advice on paper structure improvement. In our final version, we will (1) remove redundancy between the abstract, introduction, and methods, (2) separate out the lengthy motivation part of section 3 and combine it with section 2 to make the flow more natural, and (3) add figures for zero-shot and supervised settings. We also agree that the current Figure 2 might be a bit condensed and we will simplify the structure by separating extraction, construction, and propagation into a more logical sequence.
>
>
> **[== challenges in dealing with ChatGPT label inconsistencies ==]**
>
> Second, we understand the reviewer’s concern about the label inconsistencies of ChatGPT. Referring to the zero-shot experiment where ChatGPT is used to predict labels, in our prompt, we specifically constrain the labels within a range (e.g., only produce an integer score in a range, or only choose a label from a fixed set of choices), and we find that ChatGPT rarely generates novel labels beyond the predefined set. If due to fluctuation where it generates novel labels (e.g., a label number out of range) out of definition, we default the output to the most frequent label class in the training data. For value extraction from ChatGPT, similarly, we define the range in a prompt and filter out the novel labels (after canonicalizing plurals and upper/lower cases). We will elaborate on these aspects in our subsequent revision.
>
>
> **[== possibility of label leakage or user overlap between our and previous data==]**
>
> Third, we would like to note that the only difference between our data and that of (Sun et al., 2023) is network edges between users and additional influential nodes (who are followed by and not contained in the original users) added to capture the structure of the real network. There is also no annotation done on response labels. The news messages, user history, and response labels remain the same as those of the original dataset in every split. Our data is free of label leakage as we do not use any evaluation labels during training.
>
> User overlap is a scenario that also happens in the original dataset from (Sun et al., 2023) (e.g., the same user responds to message A in the training split, and message B in the test split), and we took this opportunity to assess how well the model performs under Unseen User case by removing overlapping users from the evaluation data, and we find our method works well in both overlapping and unseen user situations.
>
> We will make the points more clear in the next revision.
>
>
> **[== rationale behind the choice of belief features and future extension ==]**
>
> Finally, we choose belief features based on advice from social scientists and social psychologists on what human attributes might effectively forecast human action. We concluded with MORAL VALUES and HUMAN VALUES, which are guiding principles that help predict human reactions to social issues. For instance, previous research [1] has shown that moral values are associated with vaccination rates in the United States. In our future work, we will expand the belief graph to contain 1) fine-grained political ideologies 2) similarity-based pseudo edges between news (with the hypothesis that they make forecasting on similar news topics and isolated users more effective).
>
> > [1] Reimer, Nils Karl, et al. "Moral values predict county-level COVID-19 vaccination rates in the United States." American Psychologist 77.6 (2022): 743.

---

### Official Review · Reviewer_2MJW · 2023-08-11

**Soundness:** 4

**Excitement:**

4: Strong: This paper deepens the understanding of some phenomenon or lowers the barriers to an existing research direction.

**Paper Topic And Main Contributions:**

In this paper, the authors propose a framework called SocialSense. This framework uses ChatGPT to create a belief-centered graph leveraging profile information of users, their social content, and their historical posts. The objective of this framework is to predict responses of new articles for a given persona of users in terms of intensity and polarity of sentiments. The paper is more oriented towards real-life application of Large Language Models.


**Questions For The Authors:**

The responses from Large Language Models like ChatGPT can vary. Please share your plans to maintain consistency in the responses generated by ChatGPT.

Was any kind of human evaluation or validation done to understand if ChatGPT is suitable for unmasking latent persons?

For Table-2, could you please highlight how significant are the improvements?


**Reasons To Accept:**

The authors proposed a novel framework which leverages ChatGPT to predict responses of news articles for a given persona of users in terms of intensity and polarity of sentiments. This application of ChatGPT is innovative. It outperforms existing state-of-the-art approaches like InfoVGAE.


**Reasons To Reject:**

The authors haven't shared how they will maintain consistency in the responses generated by ChatGPT.
Effectiveness of ChatGPT for unmasking latent personas needs to be validated.


**Reproducibility:**

3: Could reproduce the results with some difficulty. The settings of parameters are underspecified or subjectively determined; the training/evaluation data are not widely available.

**Reviewer Confidence:**

3: Pretty sure, but there's a chance I missed something. Although I have a good feel for this area in general, I did not carefully check the paper's details, e.g., the math, experimental design, or novelty.

---

> ### Author Rebuttal · Authors · 2023-08-28
>
> We thank Reviewer 2MJW for the encouraging remarks on our work, especially on the innovativeness and effectiveness of our methodology. We address the comments as follows.
>
> **[== Consistency in the responses generated by ChatGPT ==]**
>
> We understand the reviewer’s concern about the consistency of the response made by ChatGPT in real applications. In our work, for example, we used belief extracted by ChatGPT as an intermediate feature to help build the network. Indeed, the belief annotation from ChatGPT alone in one run might not be the best quality, which the method depends on. One approach to ensure high quality and consistency is to aggregate results on multiple offline LLMs (e.g., LLaMA and MPT) and perform ensemble prediction.
>
>
>
>
> **[== Effectiveness of ChatGPT for unmasking latent personas ==]**
>
> We agree with the reviewer on the importance of analyzing ChatGPT’s capability of unmasking personas. In our recent progress, we sampled 50 ChatGPT extraction results done on user histories and distributed them to three human raters to verify the accuracy of extracted profiles. The human raters are graduate students who passed an initial quiz of 8 samples for filtering. We find on average the raters give a score of 3.9/5 in terms of accuracy. We will add the study to the next revision. In addition to this, in our work, we use the latent personas as intermediate nodes to enhance response forecasting instead of serving as final labels. Our empirical results demonstrate the effectiveness of the belief nodes in multiple scenarios, showing that the quality of the current annotation does have the merit of contributing to performance positively.
>
>
>
> **[== Presentation ==]**
>
> We will surely highlight the significance of improvements in the final version.

---

### Meta-Review · Area_Chair_2xih · 2023-09-14

**Recommendation:** 4

**Metareview:**

The paper advances the state of the art for response forecasting using novel methodology leveraging augmentation of social graph with LLMs. The claims are backed with strong evidence. The reviewers appreciated the novelty of the work along with the experimentation. The authors are recommended to address the reviewers' concerns in the camera ready, if the paper is accepted.

---

### Decision · Program_Chairs · 2023-10-07

**Decision:**

Accept-Main

**Comment:**

The paper advances the state of the art for response forecasting using novel methodology leveraging augmentation of social graph with LLMs. The claims are backed with strong evidence. The reviewers appreciated the novelty of the work along with the experimentation. The authors are recommended to address the reviewers' concerns in the camera ready, if the paper is accepted.